# Risk Factors for Periprosthetic Joint Infection after Shoulder Arthroplasty: Systematic Review and Meta-Analysis

**DOI:** 10.3390/jcm11144245

**Published:** 2022-07-21

**Authors:** Hyun-Gyu Seok, Jeong-Jin Park, Sam-guk Park

**Affiliations:** Department of Orthopedics, Yeungnam University Medical Center, Daegu 41000, Korea; rkaldhkthfl4@naver.com (H.-G.S.); wjdwls3912@naver.com (J.-J.P.)

**Keywords:** shoulder arthroplasty, shoulder replacement, infection, periprosthetic joint infection, risk factor, meta-analysis

## Abstract

Periprosthetic joint infection (PJI) after shoulder arthroplasty is a devastating complication that requires several additional surgeries. The purpose of this study was to assess the evidence regarding risk factors for PJI and identify those that contribute to infection by performing a meta-analysis. We searched the PubMed, Embase, and Cochrane Library databases for studies that assessed the risk factors for infection after shoulder arthroplasty. After performing screening and quality assessment on the articles, we obtained two case-control studies and six retrospective cohort studies (total of 420 infected cases and 28,464 controls). Review Manager 5.4 was used to assess the heterogeneity and odds ratio for 20 different factors that broadly included demographic factors, perioperative factors, and comorbidities. Factors that are markedly associated with PJI after shoulder arthroplasty were male sex, operation history, revision arthroplasty, acute trauma, and non-osteoarthritis as a preoperative diagnosis. Statistical analysis revealed that diabetes mellitus, liver disease, alcohol overuse, iron-deficiency anemia, and rheumatoid arthritis were risk factors for PJI after shoulder arthroplasty. The result of analysis shows that several specific factors can be targeted to prevent infections after shoulder arthroplasty. Surgeons should consider the risk factors and perform the appropriate management for patients.

## 1. Introduction

Shoulder arthroplasty is a relatively common procedure for acute trauma, such as proximal humerus fracture, and degenerative diseases such as cuff tear arthropathy, rheumatoid arthritis (RA), and osteoarthritis (OA) [1,2,3,4,5,6]. Owing to the development of devices and surgical techniques, surgical outcomes have been gradually improving, the indications for shoulder arthroplasty expanding, and the number of cases increasing [7,8,9,10]. Despite various efforts to improve the surgical outcome and to reduce complications, complications such as instability and infection after shoulder arthroplasty are still a problem [11,12]. Periprosthetic joint infection (PJI) is a devastating complication that requires readmission and several additional surgeries [13]. Therefore, it would be helpful to know which factors are related to postoperative infection. Previous epidemiological studies assessed the factors associated with PJI after shoulder arthroplasties, such as sex, age, race, body mass index (BMI), American Society of Anesthesiology (ASA) score, diabetes mellitus (DM), hypertension (HTN), operation history, and diagnosis prior to arthroplasty [14,15,16,17,18]. However, comprehensive systematic reviews of risk factors for PJI after shoulder arthroplasty are few. There was one meta-analysis of infection after shoulder and elbow arthroplasty. However, the definition of infection in the article was not clear and differed among articles. In addition, although the total number of papers included was large, the number of papers used for risk factor comparison was insufficient [8]. Thus, this study aimed to assess the evidence on risk factors for postoperative infection and identify factors that contribute to PJI by performing a meta-analysis. Demographic factors, comorbidities, preoperative characteristics of patients, and other potential factors were used for this meta-analysis.

## 2. Materials and Methods

The present systematic review was conducted in accordance with PRISMA guidelines [19].

### 2.1. Literature Search Strategy and Study Selection

We searched for relevant articles from January 2010 to December 2021 by using PubMed, Embase, and Cochrane Library to identify all published studies in which the risk factors for infection were assessed after shoulder arthroplasty. Each paper had a different definition for infection. Therefore, for accurate meta-analysis, the authors put the most effort into selecting the criteria for PJI. We considered the Musculoskeletal Infection Society (MSIS) criteria for PJI to be the most objective and useful tool. We excluded all papers that did not provide a clear definition of infection. Only when the definitions described in the paper met the MSIS criteria were they included in the analysis (Table 1).

In MSIS criteria, there are major criteria and minor criteria; PJI is present when 1 major criterion exists or 4 out of 6 minor criteria exist. The major criteria were as follows: (1) two positive periprosthetic cultures with phenotypically identical organisms; and (2) a sinus tract communicating with the joint. The minor criteria were as follows: (1) elevated serum erythrocyte sedimentation rate (ESR) and serum C-reactive protein (CRP) concentration; (2) elevated synovial leukocyte count; (3) elevated synovial neutrophil percentage; (4) presence of purulence in the affected joint; (5) isolation of a microorganism in one culture of periprosthetic tissue or fluid; and (6) greater than five neutrophils per high-power field in five high-power fields observed from histologic analysis of periprosthetic tissue at ×400 magnification.

The following search terms were used: (“factor” or “predictor” or “risk”), (“infection” or “periprosthetic infection” or “periprosthetic joint infection”), and (“shoulder arthroplasty” or “shoulder replacement”). We applied the following inclusion criteria for the selection of articles: (1) studies published in English; (2) quantitative studies, such as cohort studies or case-control studies; (3) studies with demographic, comorbid, and perioperative risk factors; and (4) studies with adequate control groups and that reported the actual numbers of patients; (5) studies that clearly describe the definition of infection and conform to the MSIS criteria [20].

The exclusion criteria were as follows: (1) case reports, reviews, or other indistinct forms; (2) studies that do not focus on risk factors; and (3) studies that repeatedly publish the same data.

We excluded all papers that did not provide a clear definition of infection. Only when the definitions described in the paper met the MSIS criteria were they included in the analysis.

**Table 1 jcm-11-04245-t001:** Definition of infection described in studies.

Authors (Year)	Definition of Infection
Diamond et al. (2021) [21]	Infection and inflammatory reaction due to internal joint prosthesis.
Florschütz et al. (2015) [22]	Positive results on a joint fluid culture, a synovial/bone tissue culture.
Morris et al. (2015) [23]	Positive results on a joint fluid culture, a synovial/bone tissue culture.
Richards et al. (2014) [24]	Purulent drainage from the deep incision, fever, localized pain or tenderness, a positive deep culture. A diagnosis of deep infection made by the operating surgeon based on intraoperative findings.
Singh et al. (2012) [25]	Positive joint fluid culture from a needle aspiration, arthroscopic procedure, fluid obtained at surgery, or fluid draining from a wound communicating with the humerus. Clinically suspected septic arthritis plus either culture-negative purulent or serosanguineous joint fluid or necrotic joint tissue (or culture not performed) or positive blood culture Frank pus/purulent material at surgery/positive synovial or bone tissue culture.
Werthel et al. (2017) [26]	Positive joint fluid culture from needle aspiration, arthroscopic procedure, fluid obtained at surgery. Fluid draining from a wound communicating with the humerus or positive synovial or bone tissue culture.
Everhart et al. (2017) [27]	ICD-9 code: osteomyelitis (730.00–730.99), septic arthritis (711.0), abscess, cellulitis (682), and infection or inflammatory reaction resulting from the joint implant or other hardware (996.66 or 996.67).
Johansson et al. (2017) [28]	Bacterial growth in more than 2 out of 5 cultures
Nezwek et al. (2021) [29]	Prosthetic shoulder infection was ultimately diagnosed using major and minor criteria updated by the Musculoskeletal Infection Society in 2011.
Nagaya et al. (2017) [30]	According to IDSA guidelines by the presence of a sinus tract communicating with the prosthesis, histopathological analyses with the presence of inflammatory cells. Visible purulence surrounding the prosthesis, and/or identical microorganisms isolated from two or more cultures.

ICD, International Classification of Diseases; IDSA, Infectious Diseases Society of America.

### 2.2. Data Extraction

Two of three authors (S.-g.P., H.-G.S.) independently evaluated the potentially eligible studies. After excluding duplicate studies, the remaining studies were screened for eligibility based on the title and abstract. Disagreements were resolved via consensus. After screening, the full texts of the eligible articles were read independently by the two authors, and the eligibility of each article was reassessed. The third author (J.-J.P.) was included in the discussion when conflicts occurred.

### 2.3. Quality Assessment

The methodological quality of the included papers was assessed using two different tools. For case-control studies, the Newcastle–Ottawa scale for case-control studies was used for the quality assessment [31]. For cohort studies, the Newcastle–Ottawa scale for cohort studies was used. The quality of each study was graded as good, fair, or poor [31]. All studies evaluated by the NOS test were confirmed to be of good quality (Table 2).

### 2.4. Factors Identified

Twenty different factors were investigated for their association with infection after shoulder arthroplasty. Demographic factors include sex, age, obesity, ASA scale, and race. Among them, age was divided into a group over 65 years old, which is generally defined as the elderly, and a group under 65 years old. Obesity is defined as a BMI of 30 or more by the World Health Organization, so it was divided into a group with a BMI of 30 or more and a group with a BMI of less than 30. The ASA scale was divided into ASA I or II, and higher groups. Finally, race was divided into two groups: white and other races. Non-arthroplasty surgery on the ipsilateral shoulder before arthroplasty, revision arthroplasty, surgery for acute trauma, and surgery for a diagnosis other than osteoarthritis were considered potential risk factors. For the type of devices, total shoulder arthroplasty including both anatomic and reverse and hemiarthroplasty was compared. In addition, the relationship between comorbidities and PJI after arthroplasty was assessed. The comorbidities used in the analysis are as follows: DM, HTN, smoking, alcohol overuse, liver disease, renal failure, iron-deficiency anemia, pulmonary disease, RA, and steroid use.

### 2.5. Statistical Analyses

RevMan 5.4 software was used for the statistical analyses of pooled data. To measure the extent of inconsistency among the results, a heterogeneity test was performed using I^2^ statistics for each analysis. An I^2^ value < 50% indicates the homogeneity of the pooled data, and the fixed-effects model was used for the analysis. On the other hand, an I^2^ value ≥ 50% indicates heterogeneity of the pooled data, and the random-effects model was used. We analyzed the odds ratios (OR) and 95% confidence intervals (CIs) to analyze dichotomous factors such as demographic factors, perioperative factors, and presence of comorbidities. A *p*-value < 0.05 was considered significant.

## 3. Results

### 3.1. Study Selection and Characteristics

Figure 1 shows the flowchart of the article screening and the detailed selection process. Of the initial 911 articles, 206 were duplicate articles and hence removed. The title and abstracts of the remaining articles were reviewed for initial screening, and 30 articles were considered appropriate for the next stage of review. After a detailed assessment, 20 articles were excluded by applying the inclusion and exclusion criteria. Finally, 10 studies were included in our meta-analysis: 8 were retrospective cohort studies and 2 were retrospective case-control studies, overall including a total of 420 cases and 28,464 controls. Table 3 shows the characteristics of all included studies.

### 3.2. Meta-Analysis Results

#### 3.2.1. Demographic Factors

Figure 2 shows the forest plots, pooled ORs, 95% CI, and heterogeneity for demographic factors.

Sex

All studies can be used to assess sex as a risk factor. Our results suggested that males were more likely to develop infection after shoulder arthroplasty than females (pooled OR = 1.71; 95% CI = 1.41, 2.09; I^2^ = 4%).

Age

Almost all of the related articles reported on age. However, the criteria for age in each study were different. Only three studies could be used for analysis. No correlations were revealed between age and PJI (pooled OR = 0.85; 95% CI = 0.60,1.21; I^2^ = 0%).

Obesity

Among the six studies that reported on BMI, three studies [21,24,25] provided detailed data for analysis. We use the international scale for BMI, which defined obesity as BMI ≥ 30 kg/m^2^. Our meta-analysis showed that the fixed-effects pooled OR for obesity compared with BMI < 30 kg/m^2^ was 0.66 (95% CI = 0.48, 0.92; I^2^ = 0%), suggesting that obesity is not a risk factor for PJI. On the contrary, patients with BMI < 30 kg/m^2^ have a higher risk of PJI than obese patients.

ASA scale

The data of three studies [24,25,30] could be used for the statistical analysis of six papers related to ASA. The ASA scale was used to assess a patient’s overall health before surgery. In the current study, we considered ASA ≥ 3 as a potential risk factor for PJI. However, no significant difference was found (pooled OR = 0.87; 95% CI = 0.20, 3.80; I^2^ = 82%).

Race

Two studies can be used for analyses with race as a factor [24,27]. We performed the analysis separately for white race and other races. No correlations were revealed between race and PJI (pooled OR = 1.74; 95% CI = 0.79, 3.82; I^2^ = 0%).

#### 3.2.2. Perioperative Factors

Figure 3 shows the forest plots, pooled OR, 95% CI, and heterogeneity for perioperative factors.

Previous operation history on shoulder (non-arthroplasty surgery)

Four studies [22,23,26,29] could be used for analyses with previous non arthroplasty surgery history. The results indicated that patients with surgical history are at more risk than the control group. The fixed-effect pooled OR was 2.40 (95% CI = 1.62, 3.54; I^2^ = 0%).

Diagnosis prior to shoulder arthroplasty

Several studies reported on the diagnosis prior to arthroplasty. In this meta-analysis, we attempted to determine whether the acute trauma-related diagnosis was a risk factor of PJI. Furthermore, we performed an analysis of OA compared with other diagnoses. In conclusion, preoperative diagnosis related to acute trauma is a risk factor for PJI (pooled OR = 1.74; 95% CI = 1.15, 2.62; I^2^ = 0%), and arthroplasty for OA has a lower risk of PJI than other diagnoses (pooled OR = 0.57; 95% CI = 0.36, 0.89; I^2^ = 0%). Five trauma-related studies [24,25,26,27,30] and three OA-related studies [22,25,26] were used for the meta-analysis.

Primary arthroplasty versus revision arthroplasty

Three studies [23,27,30] provided data for comparing primary arthroplasty versus revision arthroplasty. The random-effects pooled OR for primary surgery compared with revision surgery was 0.21 (95% CI = 0.08, 0.57) with heterogeneity (I^2^ = 58%).

Total arthroplasty versus hemiarthroplasty

Anatomic total shoulder arthroplasty (aTSA), reverse TSA (rTSA), and hemiarthroplasty (HA) are three of the most common procedures in shoulder arthroplasty. Two studies [24,26] included aTSA, rTSA, and HA. One study [22] included aTSA and rTSA only, and another study [30] included aTSA and HA only. Four studies [21,23,25,29] were recorded for only one type of operation. In the remaining three studies [27,28,30], there was no detailed description of whether TSA was anatomical or reversed. There were many papers that were not clearly classified and labeled, so the analysis was conducted first by comparing total shoulder arthroplasty (including rTSA and aTSA) and HA. The results show that the type of surgery had no correlation with PJI (pooled OR = 0.77; 95% CI = 0.53, 1.11; I^2^ = 45%) Then, aTSA versus rTSA (pooled OR = 0.61; 95% CI = 0.30, 1.23; I^2^ = 53%), aTSA versus HA, and rTSA versus HA (pooled OR = 1.14; 95% CI = 0.50, 2.60; I^2^ = 68%) were analyzed, respectively. Among these, only the analysis comparing aTSA and HA (pooled OR = 2.09; 95% CI = 1.18, 3.71; I^2^ = 0%) showed a significant result, and in the case of aTSA, the risk of PJI was higher than that of HA.

#### 3.2.3. Comorbidities

Several studies have reported comorbidities such as diabetes and metabolic syndrome as risk factors for PJI after shoulder arthroplasty [18,23,32]. In the current study, we performed a meta-analysis of comorbidities reported in at least two of the selected ten articles.

Figure 4 shows the forest plots, pooled OR, 95% CI, and heterogeneity for comorbidities with statistically significant results. DM was reported in 6 of 10 studies [21,23,24,27,29,30]. Our statistical analysis evaluated DM to be a risk factor for PJI after shoulder arthroplasty (pooled OR = 1.32; 95% CI = 1.04, 1.68; I^2^ = 0%). Furthermore, patients with liver disease have a higher risk for postoperative infection after shoulder arthroplasty than that of the control groups (pooled OR = 1.70; 95% CI = 1.18, 2.44; I^2^ = 0%). RA was also found to be a risk factor for PJI as a result of the analysis (pooled OR = 1.59; 95% CI = 1.20, 2.11; I^2^ = 26%).

Data required for the analysis of alcohol overuse (pooled OR = 2.47; 95% CI = 1.63, 3.74; I^2^ = 0%) and iron-deficiency anemia (pooled OR = 2.73, 95% CI = 1.04, 7.16, I^2^ = 71%) were provided only in two articles. Both of them were identified as risk factors for PJI as a result of the meta-analysis. By contrast, statistical analysis revealed no correlation between HTN (pooled OR = 1.04; 95% CI = 0.67, 1.60; I^2^ = 0%), smoking (pooled OR = 1.37; 95% CI = 0.74, 2.53; I^2^ = 0%), pulmonary disease (pooled OR = 1.26; 95% CI = 0.96, 1.66; I^2^ = 0%), renal failure (pooled OR = 1.96; 95% CI = 0.82, 4.65; I^2^ = 0%), or steroid use (pooled OR = 3.12; 95% CI = 0.84, 11.56; I^2^ = 52%), and PJI.

#### 3.2.4. Publication Bias

A funnel plot analysis was performed for demographic characteristics, perioperative factors, and comorbidities. In addition, Egger’s test was performed for factors that have been reported in more than three studies. The *p*-value for all factors were > 0.05. (Sex, *p* = 0.089; age, *p* = 0.3333; obesity, *p* = 0.7696; ASA scale, *p* = 0.4648; DM, *p* = 0.4137; smoking, *p* = 0.1221; pulmonary disease, *p* = 0.6876; liver disease, *p* = 0.1872; RA, *p* = 0.1146; steroid use, *p* = 0.6681; previous operative history, *p* = 0.4833; acute trauma as prior diagnosis, *p* = 0.7808; OA as prior diagnosis, *p* = 0.934; type of surgery, *p* = 0.5502; primary versus revision, *p* = 0.2703).

## 4. Discussion

Accordingly, there have been various studies [17,23] on the risk factors of postoperative infection after shoulder arthroplasty, but there is only one paper that comprehensively performed a meta-analysis and a systematic review [8]. The systematic review article analyzed postoperative infection risk factors for both shoulder and elbow arthroplasty, and definition of infection on the article was not clear. Therefore, there is no meta-analysis of PJI risk factors for shoulder arthroplasty.

Therefore, in this study, we planned a meta-analysis of postoperative infection risk factors of shoulder arthroplasty including HA, aTSA, and rTSA, which are types of shoulder arthroplasty. We broadly classified the factors into demographic factors, perioperative factors, and comorbidity.

First, among the demographic factors in the meta-analysis, the factors related to postoperative infection were sex and BMI < 30. Sex was mentioned in all included literature. In the included studies, male sex was reported as a significant risk factor in four articles, and this finding was consistent with the results of the meta-analysis [21,24,26,27]. For BMI, a patient’s high BMI is associated with infection [32]. However, in the meta-analysis performed in the current study, postoperative infection showed a significantly higher incidence rate in patients with BMI < 30. Diamond et al. Ref. [21] suggested that obesity, malnutrition, and pathologic weight loss are risk factors. The results of the existing literature are contrary to the results of the analysis. BMI is affected by variant factors. For example, Men have a higher proportion of BMI < 30 than women and have a higher proportion of patients with a BMI < 30 who have had previous surgeries. In Refs. [33,34], male and revision arthroplasty were identified as risk factors for PJI in this meta-analysis. In addition to this, other various factors can affect BMI, so it is thought that the results are different from the existing literature.

The results of meta-analysis showed that patients who underwent shoulder arthroplasty for OA have a lower PJI risk than that of patients with other diagnoses, and shoulder arthroplasty for acute trauma is a risk factor for infection. Arthroplasty associated with acute trauma, including open fracture, is more likely to cause delayed infection and osteomyelitis, than other diagnoses. In addition, compared to the control group, the difficulty of the operation and the operation time is increased in patients who underwent shoulder arthroplasty because of damage to the surrounding soft tissues when the fracture occurs. According to the contents of the paper, infection increases when the operation time increases during knee arthroplasty surgery [35]. This may have been a reason of risk for postoperative infection. The presence of previous non-arthroplasty operation history on shoulder and revision arthroplasty were both risk factors for PJI. This seems to be the cause of the discomfort of the operation, and the extension of the operation time is due to the adhesion of soft tissue from previous surgery [35]. Regarding the type of surgery, aTSA showed a high risk when comparing aTSA and HA, but the number of papers used for analysis was small. It is thought that additional study is necessary for the type of surgery.

As a result of the analysis in this study, DM, liver disease, alcohol overuse, iron-deficiency anemia, and RA were considered risk factors among comorbidities. Many studies have reported that DM is a major risk factor for PJI after orthopedic surgery [36,37]. The present study was able to collect data from included studies, and the results of the analysis showed that DM is a risk factor. In the case of iron-deficiency anemia, the correlation with blood transfusion is higher than that in anemia itself, as reported by Everhart et al. [27] This study also tried to analyze blood transfusion, but this topic was reported only in two papers. Furthermore, the data heterogeneity was severe; therefore, it was not suitable for analysis. Doran et al. [38] report that patients with RA were at increased risk of developing PJIs compared with non-RA subjects.

### Study Limitations

This study has some limitations. First, the quality of the final included studies in the meta-analysis was not high. High-quality studies, such as prospective cohort studies and randomized controlled trials, are ideal for meta-analyses, but the articles included in this study consist of two case-control studies and eight retrospective cohort design studies. Second, age and BMI were factors in a number of articles dealing with postoperative infection after shoulder arthroplasty. However, the limitation of the study was that the classification criteria for age and BMI were different for each paper; it was possible to analyze only three papers each, so the desired volume of patients could not be obtained. Lastly, a relatively low number of studies was included in our analysis. For accurate analysis, we included only ten papers in which the number of experimental groups and control groups were clearly described; therefore, a relatively small number of papers were included. Quantitative synthesis was not possible for all variables evaluated in the included studies because of the small number of studies examining some individual factors or the heterogeneity of measures.

## 5. Conclusions

The results of the meta-analysis suggested that the risk factors include male, BMI < 30, previous non-arthroplasty operation history on ipsilateral shoulders, revision arthroplasty, diagnosis prior to arthroplasty (non-OA and acute trauma), DM, liver disease, iron-deficiency anemia, alcohol overuse, and RA. Surgeons should consider the risk factors and perform the appropriate management, such as the perioperative usage of antibiotics, when planning shoulder arthroplasty for patients.

## Figures and Tables

**Figure 1 jcm-11-04245-f001:**
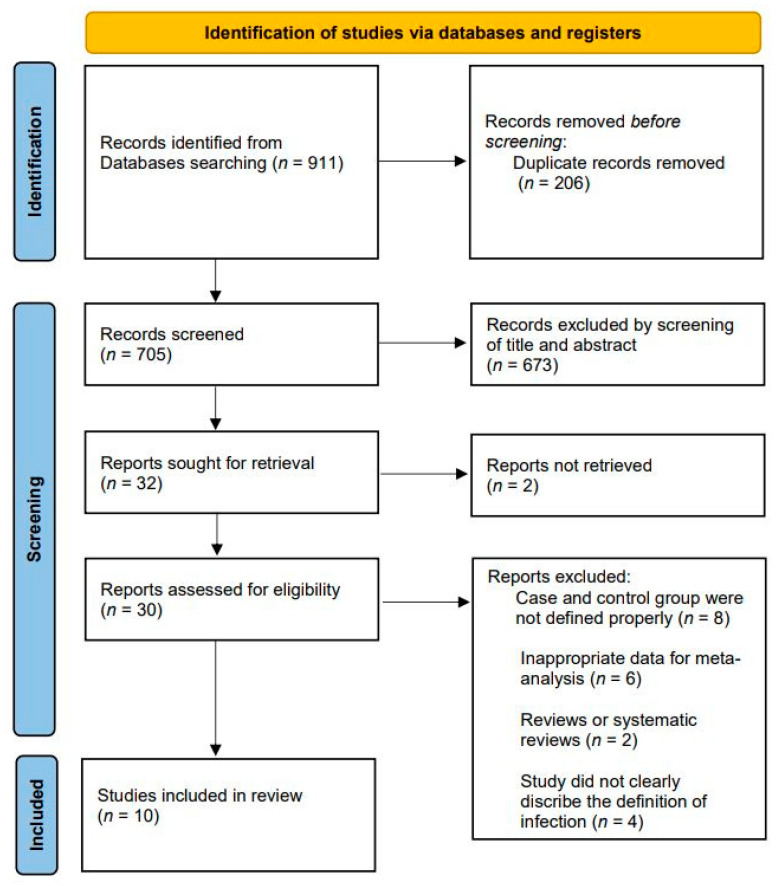
Flow chart for literature identification using Preferred Reporting Items for Systematic Reviews and Meta-Analyses (PRISMA) guidelines.

**Figure 2 jcm-11-04245-f002:**
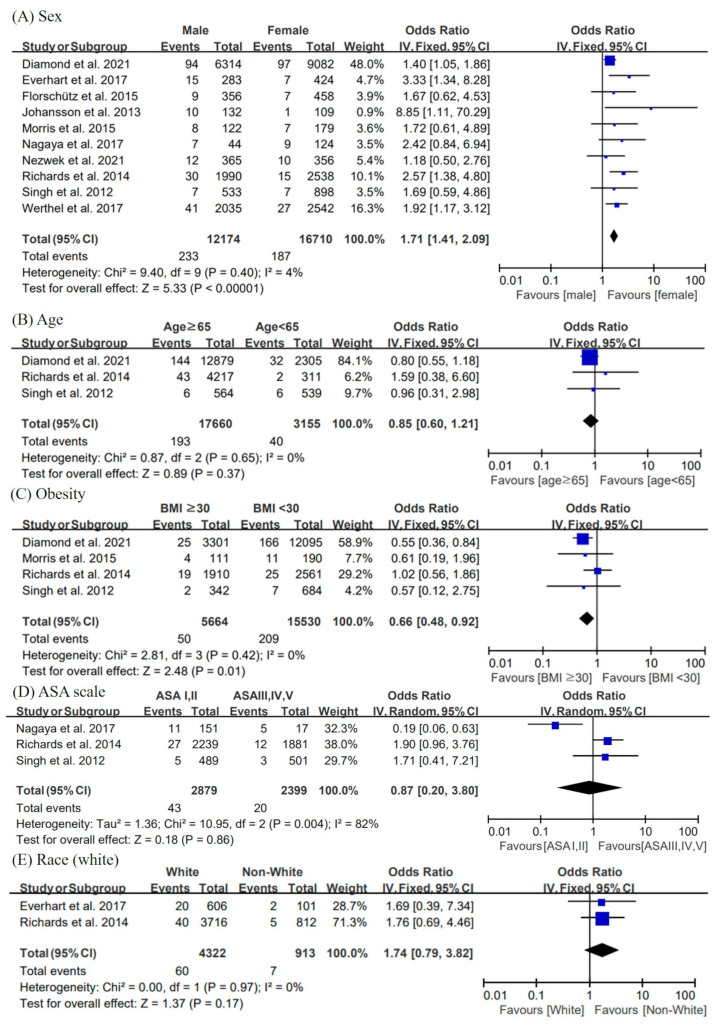
Forest plots, pooled odds ratio, 95% confidence interval, and heterogeneity for demographic factors. The risk factors analyzed in the demographic subgroups include (**A**) male (vs. female), (**B**) age (≥65 years old) (**C**) obesity (BMI ≥ 30), (**D**) ASA scale I, II versus III, IV, V), and (**E**) white race versus non-white races. BMI, body mass index; ASA, American Society of Anesthesiologists [21,22,23,24,25,26,27,28,29,30].

**Figure 3 jcm-11-04245-f003:**
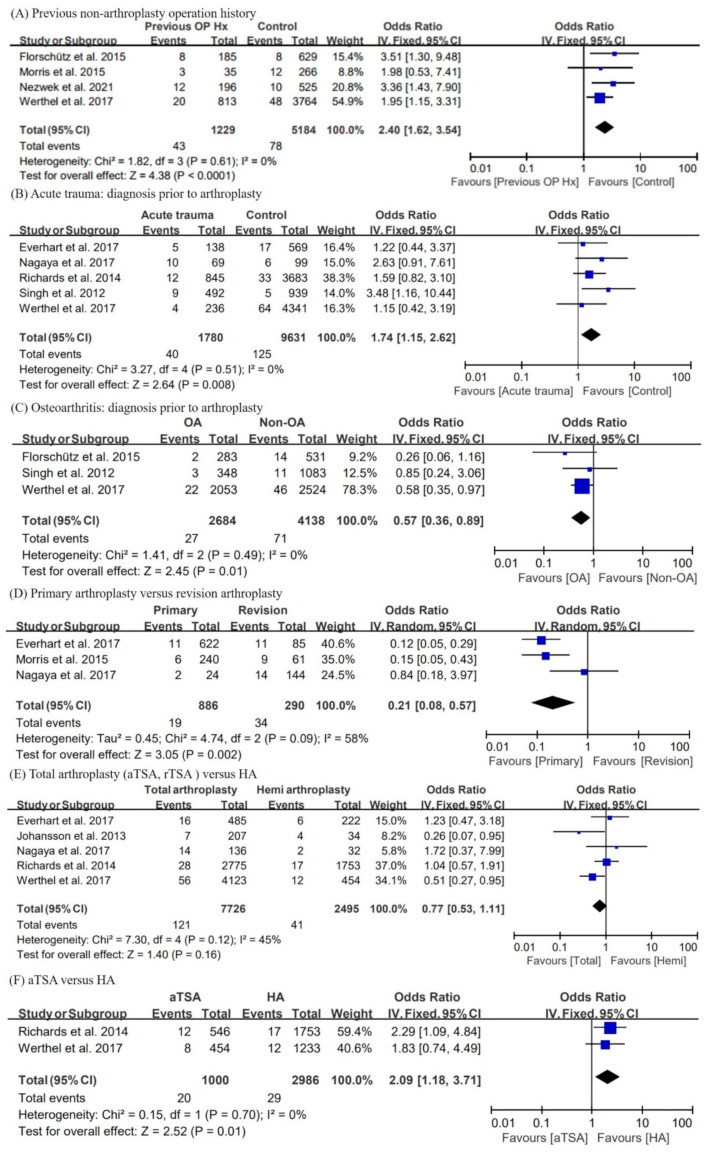
Forest plots, pooled odds ratio, 95% confidence interval, and heterogeneity for perioperative factors. The risk factors analyzed in the perioperative subgroups include (**A**) the presence of previous non-arthroplasty operation history, (**B**) acute trauma-related arthroplasty versus nontrauma-related arthroplasty, (**C**) OA as diagnosis prior to arthroplasty versus non-OA, (**D**) revision arthroplasty versus primary arthroplasty, and (**E**) total shoulder arthroplasty (aTSA and rTSA) versus hemiarthroplasty. OP Hx, operation history; OA, osteoarthritis; aTSA, anatomical total shoulder arthroplasty; rTSA, reverse total shoulder arthroplasty; HA, hemiarthroplasty [22,23,24,25,26,27,28,29,30].

**Figure 4 jcm-11-04245-f004:**
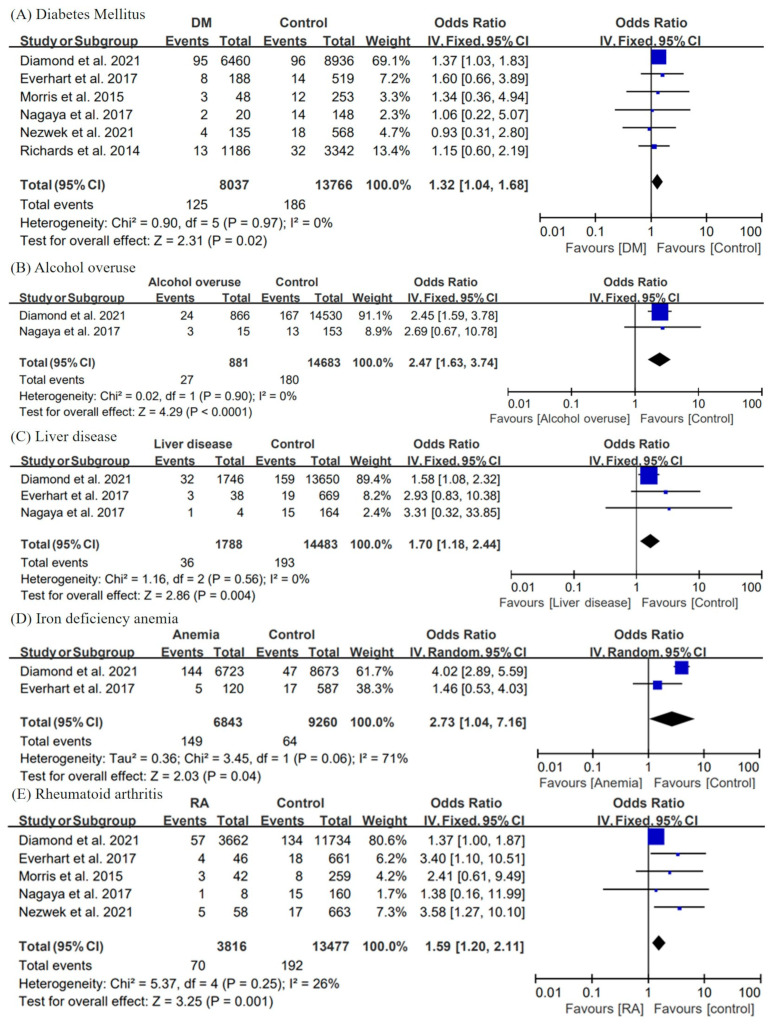
Forest plots, pooled odds ratio, 95% confidence interval, and heterogeneity for comorbidities. The risk factors analyzed in the comorbidities subgroups include the presence of previous (**A**) diabetes mellitus, (**B**) alcohol overuse, (**C**) liver disease, (**D**) iron-deficiency anemia, and (**E**) rheumatoid arthritis. DM, diabetes mellitus [21,23,24,27,29,30].

**Table 2 jcm-11-04245-t002:** Quality assessment of the studies included in this meta-analysis based on the Newcastle–Ottawa scale.

Authors	Selection				Comparability	Outcome			Total Score
	Representativeness of the exposed cohort	Selection of the non-exposed cohort	Ascertainment of exposure	Outcome of interest was not present at start		Ascertainment of outcome	Adequacy of duration of follow-up	Adequacy of completeness of follow-up	
Diamond [21]	★	★	★	★	★★	★	★		8 (good)
Florschütz [22]	★	★	★		★	★	★		6 (good)
Morris [23]	★	★	★		★★	★	★	★	8 (good)
Richards [24]	★	★	★	★	★	★	★	★	8 (good)
Singh [25]	★	★	★	★	★	★	★		7 (good)
Werthel [26]	★	★	★		★	★	★	★	7 (good)
Everhart [27]	★	★	★	★	★★	★		★	8 (good)
Johansson [28]	★	★	★		★	★	★	★	7 (good)
	Selection				Comparability	Exposure			
	Adequate definition of case	Representativeness of control	Selection of control	Definition of control		Ascertainment of outcome	Same method of ascertainment for cases and controls	Non-response rate	
Nezwek [29]	★	★	★	★	★	★	★	★	8 (good)
Nagaya [30]	★	★	★	★	★	★	★		7 (good)

**Table 3 jcm-11-04245-t003:** Characteristics of the included studies.

Authors (Year)	Study Design	Surgery Type	Infection, *n*	Non-Infection, *n*	Total, *n*	Follow Up Period	NOS Score
Diamond et al. (2021) [21]	Retrospective Cohort comparison	aTSA	191	15,205	15,396	At least 2 years (no specific number)	8 (good)
Florschütz et al. (2015) [22]	Retrospective Cohort comparison	aTSA, rTSA	16	798	814	At least 2 years (no specific number)	6 (good)
Morris et al. (2015) [23]	Retrospective Cohort comparison	rTSA	15	286	301	38.1 months	8 (good)
Richards et al. (2014) [24]	Retrospective Cohort comparison	aTSA, rTSA, HA	45	4483	4528	2.7 years	8 (good)
Singh et al. (2012) [25]	Retrospective Cohort comparison	HA	14	1417	1431	8 years	7 (good)
Werthel et al. (2017) [26]	Retrospective Cohort comparison	aTSA, rTSA, HA	68	4509	4577	63 months	7 (good)
Everhart et al. (2017) [27]	Retrospective Cohort comparison	TSA*, HA	22	685	707	30 days	8 (good)
Johansson et al. (2017) [28]	Retrospective Cohort comparison	TSA*, HA	11	230	241	2.0 years	7 (good)
Nezwek et al. (2021) [29]	Retrospective Case-Control Design	rTSA	22	699	721	18 months	8 (good)
Nagaya et al. (2017) [30]	Retrospective Case-Control Design	TSA*, HA	16	152	168	at least 2 years(no specific number)	7 (good)

TSA*: The distinction between aTSA and rTSA is not accurately described; Atsa: anatomical total shoulder arthroplasty; rTSA: reverse total shoulder arthroplasty; TSA: total shoulder arthroplasty; HA: hemiarthroplasty; NOS: Newcastle–Ottawa scale for meta-analysis.

## Data Availability

Not applicable.

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
