# Peer review of "Risk Factors for Periprosthetic Joint Infection after Shoulder Arthroplasty: Systematic Review and Meta-Analysis"

_jcm, 2022, doi:10.3390/jcm11144245_

Round 1
Reviewer 1 Report
esteemed authors, congratulations on your work,
a meta-analysis is always a demanding undertaking.
the work seems to me well conducted and well explained. Unfortunately, the quality and number of studies reported are not high
and this limits the strength of your meta-analysis
lines 269-271: the comment is misleading considering that
malnutrition and pathological weight loss can occur
(and indeed a kind of malnutrition does occur) even in obese patients
with BMI> 30
Author Response
Thank you for your sincere feedback. this is point-by-point response to comments
esteemed authors, congratulations on your work,
a meta-analysis is always a demanding undertaking.
the work seems to me well conducted and well explained. Unfortunately, the quality and number of studies reported are not high and this limits the strength of your meta-analysis
A: Thank you for your sincere feedback. As you mentioned, the authors also consider it a limitation of the journal. For accurate analysis, we included only ten papers in which the number of experimental groups and control groups were clearly described; therefore, a relatively small number of papers were included. Also, the number of high-quality studies (such as prospective cohort studies and randomized controlled trials) about PJI after shoulder arthroplasty is not enough for analysis
lines 269-271: the comment is misleading considering that
malnutrition and pathological weight loss can occur and indeed a kind of malnutrition does occur) even in obese patients with BMI> 30
A: Thanks for the good feedback. The part related to BMI was corrected due to the potential for misleading. The discussion content has been modified by referring to the feedback content.
Line 286-292 :
The results of the existing literature are contrary to the results of the analysis. BMI is affected by variant factors. For example, Men have a higher proportion of BMI < 30 than women, and have a higher proportion of patients with a BMI < 30 who have had previous surgeries. male and revision arthroplasty were identified as risk factors for PJI in this meta-analysis. In addition to this, other various factors can affect BMI, so it is thoughted that the results are different from the existing literature.
Reviewer 2 Report
This manuscript is interesting as the work is a significant contribution to the field of orthopaedics. The authors clearly described reasons of choosing those selected factors to be examined. The overall message from the article is also crystal clear; which factors are risk factors and which ones are not.
However, there are few comments for improvement:
1)I would like to suggest the authors to send the manuscript to professional English editing service as some minor mistakes (eg. unnecessary capital alphabets in line 154 and in Table 2) are noticed.
2) the inappropriate width of column in Table 2
3) Line 125- no clear difference between aTSA and rTSA. Has the authors tried to communicate and get clarification from the authors of those individual studies?
Author Response
Thank you for your sincere feedback. this is point-by-point response to comments
This manuscript is interesting as the work is a significant contribution to the field of orthopaedics. The authors clearly described reasons of choosing those selected factors to be examined. The overall message from the article is also crystal clear; which factors are risk factors and which ones are not.
However, there are few comments for improvement:
1) I would like to suggest the authors to send the manuscript to professional English editing service as some minor mistakes (eg. unnecessary capital alphabets in line 154 and in Table 2) are noticed.
A: We commissioned a professional language editing agency for English correction in order to convey clear meaning. We would like to thank Editage (www.editage.co.kr) for English language editing. We reviewed the manuscript once again to correct mistakes.
Line 154 & Table 2 : Correct Inappropriate uppercase and lowercase letters
2) the inappropriate width of column in Table 2
A: Table 2 : Correct the inappropriate width of column
3) Line 125- no clear difference between aTSA and rTSA. Has the authors tried to communicate and get clarification from the authors of those individual studies?
A: An attempt was made to contact the corresponding authors of the papers by e amil, but there was no response.

Reviewer 3 Report
This is a systematic review for risk factors of prosthetic shoulder joint infection. The authors included 10 studies and identified several factors associated with PJI.
While the goal of the study is important, I see several issues that need to be adressed:
Title: can you do a meta-analysis if the data are not coherently reported? Did you get a statistictian to look into that?
1. Introduction: Please give a more balanced introduction of shoulder TSA. Sentences such as line 32-33 are misleading. Furthermore, there is a well-written systematic review (reference 8). This should be presented in detail. Furthermore, it remains unclear where your study comes in if there already is a comprehensive review.
2. methods: as you focus on the MSIS criteria of 2011, they should be presented here. Did you require the studies included in the review to use all biomarkers given in the MSIS criteria? In your Table 1, you present a number of different definitions that do not appear to be consistent with the MSIS criteria, e.g. number of pos. cultures
Why did you not retrieve two reports?
Why did you include Everhart et al. who do not report a follow-up? This should be a standard in arthroplasty reporting.
Results are fine.
The discussion is off. Why is a BMI of less than 30 equivalent to malnutrition? There are still failry small numbers and you should discuss that in a multivariate setting patients with a normal BMI might have had a higher percentage of males or patients with previous surgeries. Those factors cannot be corrected in your review and must be discussed, particularly if findings appear illogical.
Overall, the discussion if poorly written and should be worded much more balanced and not jump to conclusion. This is a review, you do not know any of the patients and can only make assumptions. This should be reflected by the language and wording chosen. if you introduce studies in the discussion, they must be presented and weighed against your findings.
Author Response
Thank you for your sincere feedback. this is point-by-point response to comments
This is a systematic review for risk factors of prosthetic shoulder joint infection. The authors included 10 studies and identified several factors associated with PJI.
While the goal of the study is important, I see several issues that need to be adressed:
Title: can you do a meta-analysis if the data are not coherently reported? Did you get a statistictian to look into that?
A: Naturally, inconsistent data makes analysis difficult. However, due to the inherent nature of meta-analysis, I think that the numerical values and standards of each journal are inevitably different. Considering these points, we selected only journals that were deemed appropriate for rock analysis. Although we did not hire a statistician to investigate, we consulted other researchers on meta-analysis techniques and results.
- Introduction: Please give a more balanced introduction of shoulder TSA. Sentences such as line 32-33 are misleading. Furthermore, there is a well-written systematic review (reference 8). This should be presented in detail. Furthermore, it remains unclear where your study comes in if there already is a comprehensive review.
A: Thank you for your sincere feedback. In the case of the pointed-out sentence, as a result of the review, it was thought to be misleading, so it was corrected. (line 32-34). (Please see the attachment)
The Introduction part was revised by adding the contents of the mentioned articles. (line 41-44) (Please see the attachment)
To date, there is only one paper on the risk factor of PJI after shoulder arthroplasty.
Even this one paper is not only dealing with the shoulder, but a combination of shoulder and elbow.
In addition, although the total number of papers included is large, the number of papers used for risk factor comparison is insufficient. And the definition of infection in the article was not clear and differed among articles. Even superficial wound infection and deep infections were not distinguished. Considering these points, it was determined that a higher quality meta-analysis for shoulder arthroplasty PJI was necessary, and the analysis was started.
- methods: as you focus on the MSIS criteria of 2011, they should be presented here. Did you require the studies included in the review to use all biomarkers given in the MSIS criteria? In your Table 1, you present a number of different definitions that do not appear to be consistent with the MSIS criteria, e.g. number of pos. cultures
A: As you mentioned, we included the MSIS criteria of 2011 because we thought it would be helpful for readers to understand. (line 61-70) : (Please see the attachment)
Some of the articles included in this meta-analysis clearly described the definition of infection in one or two sentences, but others described it over one paragraph. Table 1 is organized by the authors as clearly as possible to help readers understand, so I think reviewers may have questions. All three authors reviewed whether it met the MSIS criteria and decided whether to include it in the analysis through discussion.
Why did you not retrieve two reports?
A: Two articles were excluded as the full text was unavailable
Why did you include Everhart et al. who do not report a follow-up? This should be a standard in arthroplasty reporting.
A: There was an error in Table 2 and it was corrected. (No data -> 30days)
(Please see the attachment)
Although the follow up period was short, Everhart et al. was judged to be a relatively well organized article on the factor of Shoulder PJI by authors. Furthermore, this paper is also included in the well written meta-analysis (reference 8: Incidence, temporal trends and potential risk factors for prosthetic joint infection after primary total shoulder and elbow replacement: Systematic review and meta-analysis) you mentioned above.
Results are fine.
The discussion is off. Why is a BMI of less than 30 equivalent to malnutrition? There are still failry small numbers and you should discuss that in a multivariate setting patients with a normal BMI might have had a higher percentage of males or patients with previous surgeries. Those factors cannot be corrected in your review and must be discussed, particularly if findings appear illogical.
Overall, the discussion if poorly written and should be worded much more balanced and not jump to conclusion. This is a review, you do not know any of the patients and can only make assumptions. This should be reflected by the language and wording chosen. if you introduce studies in the discussion, they must be presented and weighed against your findings.
A: Thanks for the good feedback. The part related to BMI was corrected due to the potential for misleading. The discussion content has been modified by referring to the feedback content.
Line 286-292 : (Please see the attachment)
The results of the existing literature are contrary to the results of the analysis. BMI is affected by variant factors. For example, Men have a higher proportion of BMI < 30 than women, and have a higher proportion of patients with a BMI < 30 who have had previous surgeries. male and revision arthroplasty were identified as risk factors for PJI in this meta-analysis. In addition to this, other various factors can affect BMI, so it is thoughted that the results are different from the existing literature.
I uploaded a revised version of the manuscript as a word file
Please see the revised manuscript

Reviewer 4 Report
It is a metanalysis and the infection is define in many version ac cording to the articoles.The markers for infections are not define and is nor presented which germs are most frecqvent.They are conclusions after other authors.
The greatest risk factors for infection after RSA were history of a prior failed arthroplasty and age younger than 65 years. Patients with these clinical characteristics should be counseled preoperatively about the increased risk for development of infection after RSA.In the articole the age is not important.
Wound complications and revision rates in patients undergoing shoulder arthroplasty who require postoperative therapeutic anticoagulation are significantly elevated and is not presented.
D vitamine severe deficiency is a another risk factor for infection.
Author Response
Thank you for your sincere feedback. this is point-by-point response to comments
It is a metanalysis and the infection is define in many version according to the articoles. The markers for infections are not define and is nor presented which germs are most frecqvent. They are conclusions after other authors.
A: The part that paid the most attention when performing this meta-analysis was the definition of infection. When reviewing more than 900 articles, few articles mention infection markers in detail, and there are articles on the causative bacteria that cause infection, but the quality is relatively low and the definition of infection is unclear. It was excluded from the analysis for reasons such as not applicable. If more papers of good quality come out, I think we can analyze them.
The greatest risk factors for infection after RSA were history of a prior failed arthroplasty and age younger than 65 years. Patients with these clinical characteristics should be counseled preoperatively about the increased risk for development of infection after RSA. In the articole the age is not important.
A: As mentioned in the results and conclusions, revision arthroplasty was explained as a risk factor for periprosthetic joint infection in this study. In the case of age, the authors expected young age to be a risk factor as mentioned in other papers, but there was no correlation as a result of the analysis. It is thought that the range of ages was different for each paper included in the analysis, and only those accurately mentioned were included in the analysis, resulting in that result.
Wound complications and revision rates in patients undergoing shoulder arthroplasty who require postoperative therapeutic anticoagulation are significantly elevated and is not presented.
D vitamine severe deficiency is a another risk factor for infection.
A: As you mentioned, risk factors for infection have been mentioned in several papers, and we aimed to conduct analysis of as many factors as possible. Several factors were actually considered, such as the need for anticoagulation, postoperative blood transfusion, immunosuppressive drugs, and psychiatric drug treatment. The number of journals dealing with the above factors was insufficient. Therefore, these factors were not included in the analysis, and if journals dealing with the factors are published in the future, analysis can be performed.
